# Can Heart Rate Variability Determine Recovery Following Distinct Strength Loadings? A Randomized Cross-Over Trial

**DOI:** 10.3390/ijerph16224353

**Published:** 2019-11-07

**Authors:** Antonia Thamm, Nils Freitag, Pedro Figueiredo, Kenji Doma, Christoph Rottensteiner, Wilhelm Bloch, Moritz Schumann

**Affiliations:** 1Department of Molecular and Cellular Sports Medicine, German Sport University, 50933 Cologne, Germany; antonia.thamm@me.com (A.T.); n.Freitag@dshs-koeln.de (N.F.); w.bloch@dshs-koeln.de (W.B.); 2Portugal Football School, Portuguese Football Federation, 1495-433 Oeiras, Portugal; pedfig@me.com; 3Research Center in Sports Sciences, Health Sciences and Human Development, CIDESD, University Institute of Maia, ISMAI, 4475-690 Maia, Portugal; 4College of Healthcare Sciences, James Cook University, Townsville 4811, Australia; kenji.doma@jcu.edu.au; 5Firstbeat Technologies, 40100 Jyväskylä, Finland; christoph.rottensteiner@firstbeat.com

**Keywords:** fatigue, recovery, MVC, HRV, RMSSD, strength performance

## Abstract

This study aimed to compare the acute effects of hypertrophic (HYP) and maximum strength (MAX) loadings on heart rate variability (HRV) and to compare possible loading-specific alterations with other markers of recovery. Ten young men with strength training experience performed two leg press loadings (HYP: five times 10 repetitions at 70% of one repetition maximum (1RM) with 2 minutes inter-set rest; MAX: 15 times one repetition at 100% of 1RM with 3 minutes inter-set rest) in a randomized order. The root mean square of successive differences statistically decreased after both protocols (HYP: 65.7 ± 26.6 ms to 23.9 ± 18.7 ms, *p* = 0.026; MAX: 77.7 ± 37.0 ms to 55.3 ± 22.3 ms, *p* = 0.049), while the frequency domains of HRV remained statistically unaltered. The low frequency (LF) band statistically increased at 48h post-MAX only (*p* = 0.033). Maximal isometric voluntary contraction (MVC) statistically decreased after HYP (*p* = 0.026) and returned to baseline after 24h of recovery. Creatine kinase (CK) statistically increased above baseline at 1h post-loadings (HYP *p* = 0.028; MAX *p* = 0.020), returning to baseline at 24h post. Our findings indicate no distinct associations between changes in HRV and MVC or CK.

## 1. Introduction

During strength training programming, several monitoring strategies may be employed to manage recovery, while optimizing training quality [1,2]. These include but are not limited to ratings of perceived exertion [3,4], measures of neural fatigue (e.g., dynamic and isometric strength testing and electromyography) [5,6], as well as blood-borne biomarkers (e.g., creatine kinase (CK) and myoglobin) [7,8]. However, the use of most of these methods is not feasible for the daily training practice and, thus, less labor-intensive measures are required.

Various studies have attempted to investigate autonomic modulations as an indicator of strength loading-induced stress and concomitant needs for recovery [9,10,11,12,13,14]. Strenuous strength loading stimulates the sympathetic branches of the autonomic nervous system, leading to an acute increase in systolic blood pressure and heart rate [15]. Increased heart rate, in turn, is associated with reductions in beat-to-beat variations (i.e., heart rate variability (HRV)) [16]. Following strength loading cessation, the systolic blood pressure may decrease due to reductions in cardiac output, likely due to an increase in afterload and/or a reduction in pre-load [17]. Reductions in cardiac output are again associated with an elevation in heart rate, consequently leading to a reduction in HRV. Therefore, hemodynamic changes in response to strenuous exercise may provide feedback on the function of the autonomic nervous system.

To date, only a few studies have investigated the acute effects of strength loading characteristics on HRV. Following a whole-body strength loading with varying volume (i.e., one, three, or five sets of each exercise), the indices of HRV appeared to follow a dose-response relationship, where no changes were observed after one or three sets of strength exercise, but large alterations were induced when the volume was increased to five sets per muscle group [9]. Similar findings were also observed in a squat protocol consisting of three sets of eight repetitions at 80% of one repetition maximum (1RM) as compared to only three sets of four repetitions with the same intensity [10]. These findings indicate that HRV seems to acutely reflect strength training volume beyond a certain threshold. However, in a study of Lima et al. 70% but not 50% of the 1RM led to significant reductions in HRV, up to 60 minutes post-exercise [14]. Thus, it is likely that HRV is affected by both the volume and the intensity of a given strength loading, which is also reflected in training dose-related long-term adaptations in autonomic nervous function [18]. 

A similar pattern may also be observed when comparing distinct training protocols, aiming at muscle hypertrophy or maximal strength, typically performed in both recreational and elite sport settings. In a study by Walker et al. it was shown that the origin of fatigue differs between maximal (MAX; 15 times one repetition, 3 minutes inter-set rest) and hypertrophic (HYP; five times 10 repetitions, 2 minutes inter-set rest) type of strength loadings [19]. While fatigue induced by maximal strength protocols appears to be caused by impaired neural drive, fatigue in hypertrophic strength protocols is related to peripheral changes in muscle activity [19]. In addition, vascular resistance is increased as a result of greater exercise intensity [20], and thus, it is likely that this will be more pronounced in response to maximal strength loadings. Since both vascular resistance and HRV are governed by the autonomic nervous system, the training mode may also affect HRV but this remains to be investigated.

While previous studies provided some evidence for HRV being sensitive enough to quantify physiological strain following distinct strength loadings, the associations with other markers of recovery remain mostly unknown. In a recent study it was shown that a strenuous whole body strength training protocol (six sets to failure with 90% of 10RM) led to distinct time frames for recovery of HRV, neuromuscular performance, and perceived exertion, respectively. However, no associations between these different recovery markers were observed [21]. Therefore, the aim of the present study was to expand on previous findings by assessing whether the type of exercise loading (i.e., MAX and HYP) acutely affects HRV assessed immediately after, as well as up to 48 hours following each protocol [19]. In addition, we aimed to assess whether changes in HRV induced by these distinct loading protocols are associated with changes in other markers of recovery, such as neuromuscular performance (i.e., changes in maximal and rapid force production), muscle pain, and CK. The findings of this study are crucial to further enhance our understanding of different monitoring strategies to manage an optimal training load in strength training programming. 

## 2. Material and Methods

### 2.1. Study Design

The study was performed with a randomized cross-over design. Initially, subjects were familiarized with the strength training protocol as well as the testing procedures and performed a 1RM testing. At least 48 hours later, all subjects performed two strength loading protocols in a randomized and counter-balanced order, separated by seven days. The two sessions consisted of a protocol focusing on muscle hypertrophy or maximal strength, respectively, and were performed on a leg press device. HR and HRV were assessed at the following time points: Prior to (pre) as well as immediately after (post0) each strength loading as well as 30 minutes post (post30), 1 hour post (post1h), 24 hours post (post24h), and 48 hours post (post48h) each intervention. In addition, isometric maximal voluntary contraction (MVC), rate of force development (RFD), muscle pain, and serum concentrations of CK were assessed prior to, immediately after, as well as 24 and 48 hours after each strength training protocol. To control for possible diurnal variations, all tests were performed at the same time of day (7:00–9:00 a.m.). Subjects were asked to maintain a similar nutritional intake prior to the loading and recovery measurements. Alcohol and caffeine intake were restricted for 12 h prior to the intervention protocols and throughout the two-day recovery period. Similarly, subjects refrained from strenuous physical activity for at least 48 h prior to each of the intervention protocol and recovery measurements. 

### 2.2. Subjects

Ten healthy men (Mean ± SD: Age 24 ± 3.8 years, 186 ± 7.7 cm, and 79 ± 7.6 kg) volunteered to participate in the study. Subjects were physically active and reported previous strength training experience with at least one to three weekly sessions over the six months prior to inclusion into the study. All subjects were free of acute or chronic diseases that would contraindicate the performance of heavy strength training and provided written informed consent prior to the commencement of the study. The study was conducted according to the declaration of Helsinki and received approval by the local ethics committee (087/2017).

### 2.3. Procedures

#### 2.3.1. Heart Rate and Heart Rate Variability

The autonomic modulation of the nervous system was indirectly measured through HR and HRV, with two non-invasive ECG electrodes placed on the chest (i.e., on the right side of the body right under the collarbone and on the left side of the body on the rib cage). ECG signals were collected at a rate of 1000 Hz using the Firstbeat Bodyguard 2 (Firstbeat Technologies, Jyväskylä, Finland). Resting HRV was measured in supine for 10 minutes at pre (i.e., prior to the warm-up), post0, post30, and post1h. In addition, medium-term recovery was assessed by comparing HRV determined during an orthostatic test performed in the morning of each loading day as well as at post24h and post48h. Orthostatic tests consisted of 5 minutes in supine and standing, respectively. The subjects carried out orthostatic tests independently immediately after awakening. 

For data analysis, standardized 2-minute intervals were cut from all data points and manually checked for artifacts with *Artiifact Version2* [22], supported by internal algorithms. Data with more than 10% of identified artifacts was excluded from the analysis. Removed Artifacts were not replaced. The HRV data were analyzed by *Artiifact Version2* [22] in both frequency and time domains. For the frequency domain, low-frequency (LF; 0.04–0.15 Hz) and high-frequency (HF; 0,15–0,4 Hz) bands were assessed and the LF/HF ratio was calculated. Time domain measures included the root mean square of successive differences (RMSSD). 

#### 2.3.2. Dynamic Muscle Strength 

Dynamic muscle strength in the leg press was assessed via a bilateral one repetition maximum test. Prior to the test, a short standardized warm-up was performed (one times 10 repetitions at 70% of estimated 1RM, one times seven repetitions at 75% of estimated 1RM, one times five repetitions at 80% of estimated 1RM, and one times one repetition at 90% of estimated 1RM) [19]. Thereafter, participants performed single repetitions, which progressively increased in loads (5–10 kg increments) until failure. The 1RM was set as the weight of the last successful repetition at an accuracy of 5 kg, and was to be achieved after a maximum of five trials. Inter-set rest intervals of 1 minute were provided between subsequent trials. The starting knee angle was set to 80°.

#### 2.3.3. Maximal Isometric Voluntary Contractions 

Maximal isometric voluntary contractions (MVC) and rate of force development during the first 200 ms (RFD) were measured as indices of neuromuscular fatigue for knee extensors during isometric leg press. The knee angle was set to 120° for all isometric testing. Subjects were instructed to push as hard and as fast as possible against the support and maintain maximal tension for about 3 seconds. For measuring the maximum strength ability at rest, inter-set rest periods of 1 minute were allowed during pre, post1h post24h, and post48h, while immediately after the loading inter-set rest periods of only 10 seconds were allowed, in order to assess the acute neuromuscular fatigue. A minimum of three trials was performed and additional one to two trials were added if the maximum force of the last trial differed by more than 5% from the previous trial. The trial with the highest peak force was used for statistical analysis.

#### 2.3.4. Blood Sampling

Capillary blood samples were drawn from the earlobe, to determine blood lactate concentrations throughout each loading. Capillary blood sampling was performed following each set during HYP and every third set during MAX, respectively (T1, T2, T3, T4, and T5). Capillary blood was sampled in capillary tubes (20 µL) and blood lactate concentrations were analyzed by a Biosen S-Line Analyzer (EKF Diagnostic GmbH, Barleben, Germany). In addition, venous blood samples were drawn at the following time points: Pre, post0, post1h, post24h, and post48h. Serum CK was assessed by an enzymatic kinetic assay method (Roche Diagnostics, Mannheim, Germany) using a Hitachi 912 Automatic Analyzer (Roche Diagnostics). 

#### 2.3.5. Muscle Pain

The perceived sensation of muscle pain was assessed by a 100 mm visual analogue scale (VAS) at pre, post0, post1h, post24h, and post48h. Each subject freely set a mark on the scale reaching from no to very strong pain. The measured distance from zero was used for statistical analysis.

### 2.4. Acute Loading Protocol

Prior to loading, a warm-up was carried out on a bike ergometer for 5 minutes at a self-selected intensity, followed by a set of 10 repetitions on the leg press (Gym80 International GmbH, Gelsenkirchen, Germany) at 70% of 1 RM. The HYP protocol consisted of five sets of 10 repetitions at 80% of 1RM, with 2 minutes of inter-set rest periods. The MAX protocol comprised of 15 sets of one repetition at 100% of 1RM, with 3 minutes of inter-set rest periods [19]. During the inter-set rest periods, the subjects were asked to stay seated. Verbal encouragements were given throughout all sets. The intensity for both loadings was calculated based on previously determined 1RM, albeit this was only used to determine the intensity for the first set. If the subject completed this set successfully, the load for the subsequent sets was increased by 5 kg. Likewise, if the initial set was not completed without assistance the load was reduced by 5 kg for all remaining sets. The rating of perceived exertion (from six to 20, Borg-Scale) was collected after each set to confirm whether each strength training protocol was completed at maximum effort. 

### 2.5. Statistical Analysis

All data are expressed as mean ± standard deviation (SD). Normality of data was assessed by the Shapiro Wilk Test. Data were analyzed using absolute values and are presented as Δ%, unless indicated differently. MVC and RFD were analyzed using a two-way repeated measures analysis of variance (ANOVA) to assess differences over time and between loadings. Adjustments were made with the Bonferroni post-hoc test. Given that HRV, VAS, blood lactate, and serum CK data were not normally distributed after log transformation, these measures were analyzed by a Wilcoxon signed-rank test. Bonferroni correction was applied by multiplying the *p*-values with the number of comparisons. Associations between dependent variables were assessed by Spearman rank correlation coefficients. The significance for all tests was set at *p* ≤ 0.05, while values ≤ 0.06 were accepted as a significant trend. In order to determine the magnitude of differences between time points, the effect sizes (Cohen’s d; ES) were also calculated, where d = 0.2 was considered a small ES, *d* = 0.5 a moderate ES, and d ≥ 0.8 a large ES. The post-hoc power analysis revealed a statistical power of 0.80–0.95 for measures of HRV, MVC, RFD, muscle pain, and CK.

## 3. Results

### 3.1. Heart Rate

In HYP, HR statistically increased immediately after the loading (57 ± 6 bpm to 76 ± 11 bpm, *p* = 0.001, *d* = 2.1). Within 30 minutes post exercise, HR returned to baseline values (*p* > 0.06; post30: *d* = 0.9; post1h: *d* = 0.7; post24h: *d* = −0.8; and post48h: *d* = −0.7). In MAX, HR remained statistically unaltered throughout the loading (56 ± 6 bpm to 61 ± 6 bpm, all *p* > 0.06; post0: *d* = 0.8; post30 *d* = 0.0; post1h: *d* = −0.3; post24h: *d* = 0.0; and post48h: *d* = −0.2).

### 3.2. Heart Rate Variability

RMSSD statistically decreased immediately following both protocols (HYP: *p* = 0.026, *d* = 1.0; MAX: *p* = 0.049, *d* = 1.1) (Figure 1, Table 1). Within 30 minutes post-exercise, RMSSD returned to the baseline values (*p* > 0.06). All other HRV parameters (LF, HF, and LF/HF) remained statistically unaltered during the acute recovery up to post1h (*p* > 0.06).

During recovery, LF statistically increased at post48h following MAX (*p* = 0.033, *d* = 2.1). A statistically significant between-loading difference was observed in LF at post48h (*p* = 0.01, *d* = 1.1) (Figure 2), with moderate to large between protocol effects (Table 1). All other HRV parameters remained statistically unaltered during the orthostatic tests. 

### 3.3. MVC and RFD

MVC statistically decreased immediately following HYP (*p* = 0.026, *d* = 1.3) (Figure 3), gradually returning to baseline values during the following 24 hours (post1h: *p* = 0.016, *d* = 1.4; post24h: *p* > 0.05, *d* = 0.8; and post48h: *p* > 0.06, *d* = 0.4). In MAX, MVC remained statistically unaltered throughout the loading and recovery (all *p* > 0.06; post0: *d* = 1.4; post1h: *d* = 1.1; post24h: *d* = 0.5; and post48h: *d* = 0.2). RFD remained statistically unaltered throughout the loadings and recovery (HYP and MAX all *p* > 0.06) (HYP: post0: *d* = 0.6; post1h: *d* = 0.5; post24h: *d* = 0.2; MAX: post0: *d* = 0.8; post1h: *d* = 1.0; post24h: *d* = 0.7; and post48h: *d* = 0.4). The changes in MVC and RFD from pre to post0h across both conditions were statistically associated with changes in RMSSD, when all data were pooled (r = 0.433, *p* = 0.056 and r = 0.550, *p* = 0.012).

### 3.4. Creatine Kinase

Serum CK (Figure 4) statistically increased after 1 h following both protocols (HYP: post1h: *p* < 0.028, *d* = 0.2; MAX: post1h: *p* < 0.02 *d* = 0.3) but was no longer statistically increased at 24 hours post-loading (HYP: post24h: *p* > 0.06, *d* = 0.5; post48h: *p* > 0.06, *d* = −0.1; MAX: post24h: *p* > 0.06, *d* = 0.5; and post48h: *p* > 0.06, *d* = 0.4). At 24 hours post exercise, CK was no longer statistically increased.

### 3.5. Blood Lactate and Muscle Pain

Blood lactate concentrations (Figure 5) statistically increased during HYP (T2: *p* < 0.025, *d = 1.7*; T3: *p* < 0.025, *d = 2.2*; T4: *p* < 0.025, *d* = 2.4; and T5: *p* < 0.025, *d* = 2.1) but remained statistically unaltered during MAX. 

The subjective sensation of muscle pain statistically increased immediately following both protocols (HYP: *p* < 0.02, *d* = 1.8; MAX: *p* < 0.02, *d* = 1.43), and remained statistically increased in HYP for up to 48 h post (post1h *p* < 0.02, *d* = 1.3; post24h *p* < 0.02, *d* = 1.6; and post48 *p* < 0.03, *d* = 1.5) but not MAX.

## 4. Discussion

The primary aim of this study was to evaluate the acute effects of a hypertrophic and a maximum strength training session on HRV. Furthermore, we aimed to investigate whether changes in HRV are associated with changes in other markers of recovery following the two distinct strength loadings. The main finding was that RMSSD acutely decreased to a similar magnitude following both MAX and HYP but returned to baseline within 30 minutes post-exercise. Although the magnitude of acute reductions in MVC and RFD was similar in the two loadings, statistically significant reductions were observed for MVC in HYP only. The changes in MVC and RFD from pre to post0h across both conditions were statistically associated with changes in RMSSD. In addition, while the frequency domains of HRV remained statistically unaltered during the first hour post-exercise, a statistical increase in LF was observed in MAX after 48h. 

In the present study, both strength protocols acutely reduced the vagal activity as reflected by decreases in RMSSD. These findings are well in line with previous studies [9,10,13]. However, in most of these studies reductions in vagal activity were only observed with a higher as compared to a lower training volume. For example, Figueiredo et al. showed a decrease in RMSSD for 30 minutes post-exercise after a hypertrophic loading with strength training experienced young men, when at least three sets with eight–10 repetitions of whole body strength training were performed [9]. Similarly, González-Bandillo et al. found significant reductions in RMSSD following three sets of eight repetitions back squat loading but not during a lower volume of three sets with four repetitions [10]. In line with our findings, the comparable measures between the HYP and MAX protocols suggest that repeated activation of large muscle mass with heavy loads are likely to induce acute inhibition of sympathetic nervous function. 

Despite training volume, also an exercise intensity threshold seems to exist, that needs to be exceeded to observe changes in the autonomic modulation following a strength training session. For example, Lima et al. identified that an exercise protocol (three sets with six, nine, and 12 repetitions, respectively) conducted with an intensity of 70% of the 1RM increased sympathetic as well as parasympathetic indices of HRV, while the same protocol performed with 50% of the 1RM did not induce changes in HRV [14]. Interestingly, in our study no differences between the loadings performed with 10RM or 1RM were observed. However, it needs to be noted that not only the loading intensity but also the rest intervals differed between the two protocols. While the changes in vascular resistance and, thus, the effects on HRV, are likely to be larger with heavier loads [20], it is unclear as to whether the longer rest periods during maximal strength loadings are compensating for this effect.

The shown dependence of reductions in HRV on exercise intensity and/or volume has previously been associated with an accumulation of hydrogen ions and inorganic phosphate [23]. Thus, it is likely that high levels of blood lactate and/or ammonia as well as a concomitant depletion in glycogen and phospho-creatine typically observed during heavy strength training [24] may induce reductions in HRV. However, a significant blood lactate accumulation in the present study was observed in HYP only, while RMSSD was acutely reduced both in HYP and MAX. Thus, it remains questionable, whether an accumulation of metabolites may be considered the main cause for acute reductions. Instead, the breathing patterns may also affect the HRV responses. Lifting heavy loads is often accompanied by a Valsalva maneuver [12]. This, in turn, will further increase blood pressure, ultimately affecting the HRV response [25,26]. Therefore, it is possible that mechanisms by which acute reductions in RMSSD are induced may differ between MAX and HYP.

A prominent feature of our study was the prolonged recovery period, in which HRV was assessed. Nonetheless, it appeared that most parameters returned to baseline already within 30 minutes post-exercise. This finding is in line with a previous study which did not report any prolonged changes in HRV beyond 5–10 minutes after exercise completion [10]. In contrast, MVC remained significantly decreased at 1 h post-loading, returning to baseline values 24 h post both protocols. This development in MVC ability is well in line with previous findings where a protocol quite similar to our HYP protocol (four times 12 at 100% of the 12RM squat) in strength training-experienced subjects was used [27]. As such, it appears that our subjects were recovered already quite soon following the strenuous loadings. In fact, this was also shown by the relatively low CK concentrations, despite subjective muscle pain remaining above baseline during the 48 h recovery after HYP. Moreover, despite significant associations between the pooled reductions in RMSSD and muscle force (i.e., MVC and RFD), our data indicate that different markers of recovery represent restoration of distinct systems (i.e., neuromuscular versus autonomic function) at different time points and, thus, may not be used interchangeably. 

Interestingly, a notable significant increase in LF was observed at 48h following MAX but not HYP. A similar phenomenon has previously been shown throughout 24 hours, following a 2 hour weightlifting program with experienced weightlifters (three repetitions at 60, 70, and 80% of the 1RM, two repetitions at 90 %, and one repetition at 95% with 90 seconds of rest between each repetition) (Chen et al., 2011). Based on the effect size, this increase in LF was somewhat similar to the increase of CK but was not reflected in reduced neuromuscular performance. However, when interpreting these findings, it should be noted that the implications of LF for the autonomic modulation of HRV is not yet entirely understood and it can be assumed that LF reflects both a sympathetic as well as parasympathetic activity [28]. Thus, these findings require further investigations.

Importantly, based on the effect size, the magnitude of muscle pain was actually higher in HYP compared to MAX throughout the 48 h of the recovery period. Together with our findings of increased LF in MAX, these findings provide a discrepancy between the objective measures of neuromuscular fatigue and the subjective sensation of pain, which requires further investigation. In contrast to Chen et al. who reported that HRV, as a surrogate of autonomic nervous function, may accurately mirror recovery status following strength loadings in weightlifters [29], we can currently not support the use of HRV to accurately quantify recovery demands following distinct strength-loading patterns. Such a conclusion is in line with other recent findings showing no advantage in strength gains and muscle hypertrophy when strength training was performed with a predetermined protocol or program based on HRV recovery [30]. However, caution should be paid when interpreting our present findings as the low sample size somewhat limits the generalizability of our findings.

## 5. Conclusions

Our data showed that RMSSD was reduced after a maximal and hypertrophic strength protocol, irrespective of the type of loading. Furthermore, all HRV indices initially returned to baseline within 30 minutes post-exercise but a statistical increase in LF was observed at 48 hours following the maximal strength loading only. Our findings indicate that HRV alone may not be sensitive enough to determine the recovery demands following distinct strength loadings. Furthermore, since no associations between changes in HRV and other common markers of recovery (i.e., force production, subjective wellbeing, and serum CK) were observed, practitioners are advised to incorporate multiple markers to assess recovery needs to manage strength training programming. Future studies should replicate these findings using larger sample sizes, possibly including subjects with different training backgrounds. 

## Figures and Tables

**Figure 1 ijerph-16-04353-f001:**
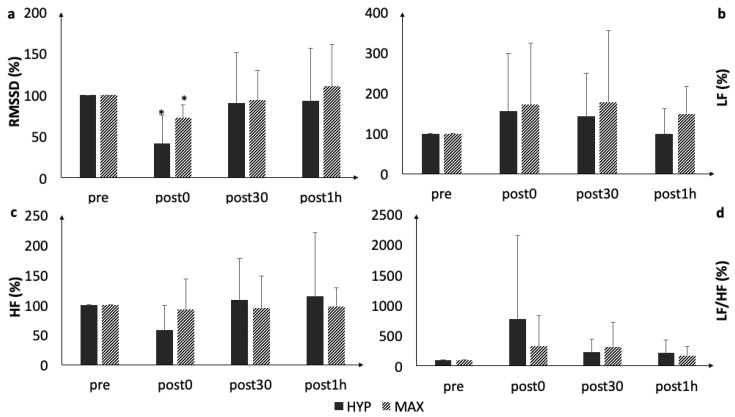
Acute changes in (**a**) root mean square of successive differences (RMSSD), (**b**) low Frequency (LF), (**c**) high Frequency (HF), and (**d**) LF/HF. *, statistically significant difference to baseline (*p* < 0.05).

**Figure 2 ijerph-16-04353-f002:**
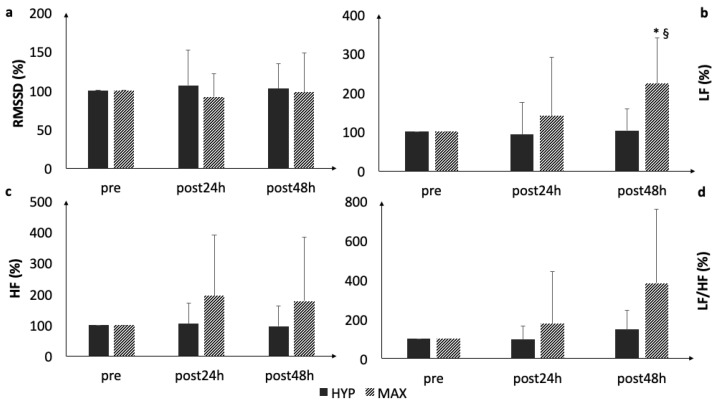
Comparison of (**a**) RMSSD, (**b**) LF, (**c**) HF, and (**d**) LF/HF as measured during the orthostatic tests. * statistically significant difference to baseline (*p* < 0.05); §, statistically significant group difference.

**Figure 3 ijerph-16-04353-f003:**
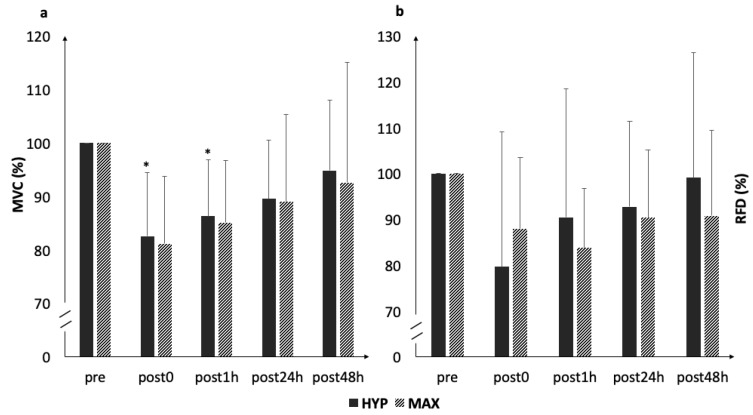
Acute changes in (**a**) maximum isometric force and (**b**) rate of force development. * statistically significant difference to baseline (*p* < 0.05).

**Figure 4 ijerph-16-04353-f004:**
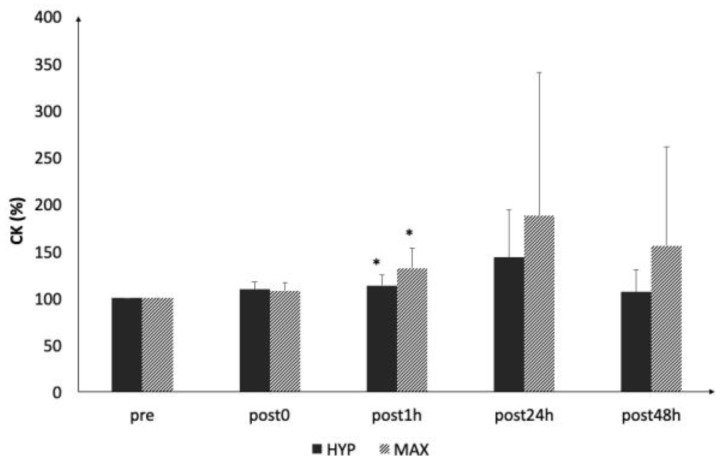
Acute changes in serum creatine kinase (CK) concentrations. * statistically significant difference to baseline (*p* < 0.05).

**Figure 5 ijerph-16-04353-f005:**
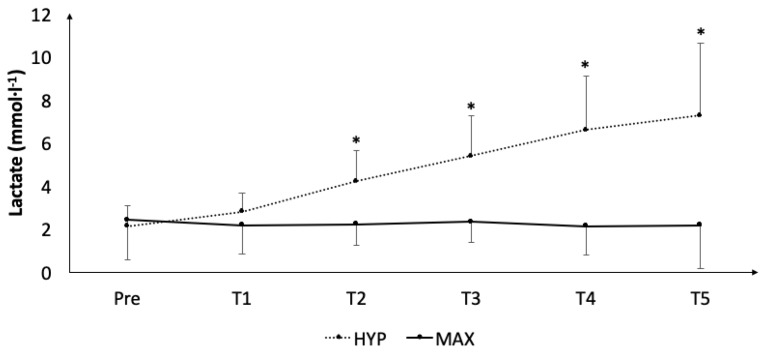
Acute accumulation of blood lactate concentrations throughout both protocols. *, statistically significant difference to baseline (*p* < 0.05).

**Table 1 ijerph-16-04353-t001:** Effect sizes (ES) between MAX and HYP for each acute time point in relation to the corresponding pre value.

Marker	post0	post30	post1h	post24h	post48h
ES (95% CI)					
RMSSD	**1.20 **** (0.20, 2.09)	0.07 (−0.81, 0.94)	0.31 (−0.58, 1.18)	−0.38 (−1.25, 0.52)	−0.11 (−0.98, 0.77)
LF	0.12 (−0.77, 0.99)	0.24 (−0.65, 1.11)	**0.76 *** (−0.18, 1.63)	0.40 (−0.50, 1.26)	**1.32 **** (0.3, 2.22)
HF	**0.73 *** (−0.21, 1.60)	−0.23 (−1.09, 0.66)	−0.22 (−1.09, 0.67)	0.62 (−0.30, 1.49)	**0.53 *** (−0.38, 1.40)
LF/HF	−0.43 (−1.30, 0.47)	0.28 (−0.61, 1.15)	−0.23 (−1.10, 0.66)	0.42 −0.49, 1.28)	**0.85 **** (−0.10, 1.73)
MVC	−0.12 (−0.99, 0.76)	n.a.	−0.11 (−0.98, 0.77)	−0.04 (−0.92, 0.84)	−0.12 (−1.00, 0.76)
RFD	0.34 (−0.56, 1.21)	n.a.	−0.28 (−1.15, 0.61)	- 0.17 (−1.04, 0.71)	−0.36 (−1.22, 0.54)
CK	−0.18 (−1.05, 0.70)	n.a.	**1.09 **** (0.11, 1.97	0.39 (−0.51, 1.52)	**0.65 *** (−0.28, 1.52)
VAS	**−0.74 *** (−1.61, 0.20)	n.a.	**−0.60 *** (−1.47, 0.32)	**−0.87 **** (−1.75, 0.08)	**−0.93 **** (−1.81, 0.03)

bold digits denoting moderate to large ES. * moderate ES; ** large ES. RMSSD: Root mean square of successive differences; LF: Low-frequency; HF: High-frequency; LF/HF: Low frequency to high frequency ratio; MVC: Maximal isometric voluntary contraction; RFD: Rate of force development; CK: Creatine kinase; and VAS: Visual analog scale.

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
