# Peer review of "Can Heart Rate Variability Determine Recovery Following Distinct Strength Loadings? A Randomized Cross-Over Trial"

_ijerph, 2019, doi:10.3390/ijerph16224353_

Round 1

Reviewer 1 Report

Major points

There are some major concerns to be addressed before I could recommend publishing:

There are numerous errors of “Tense and Grammar” throughout the manuscript (INTRODUCTION and DISCUSSION). Therefore, diligent editing is in order to fix the ENGLISH language. Also, there were too long paragraph sentences (INTRODUCTION and DISCUSSION) INSUFFICIENT literature review in the introduction and discussion sections.

Minor points

Line 28: The term "Performance" not specialized

Line 101: “Firstbeat Bodyguard 2” is reliable ?. Please, cite some published papers regarding the reliability and validity of this tool.

Line 117 - 118: How the authors selected this warm-up protocol method? Have the authors calculated the loads based on the strength training profile of participants?. Please refer to the references that modified the protocol.

Line 175- 179: The representative results in this paragraph must be formulated simply again.

Reviewer 2 Report

The introduction and discussion section should be a little more theory-driven. Although the introduction of the paper includes information concerning the effects of exercise loading on HRV and how it has been evaluated in empirical studies, little rationale is given to why this study is important for future research and practical implications. I suggest authors also to explore more the conceptualization and association among IV and DV.

The sample size seems shortcoming, thus results should be interpreted with caution. Although the statistical analyses are well described, limitations should be presented. Can the findings be generalized to all active exercisers? Can the results of this study with a sample size of 10 be used to extend previous literature? The authors are encouraged to give some more thoughtful consideration about what the study was unable to address (or addressed sub-optimally) and how these shortcomings can be addressed in future research.

Furthermore, there is insufficient explanation of the theoretical and practical consequences of the findings, or of the specific ways in which they contribute to the literature on the topic. I think the paper could benefit if, in the discussion, the meaning of the results is explaining a further through concrete examples.

Round 2

Reviewer 2 Report

I would like to commend the authors on the revision of the manuscript and thank them for the point-by-point response. The paper is greatly improved. There are a couple of minor points that could do with further clarification.

The manuscript should be reviewed by a native English speaker to check for syntax and flow (Example Line 45-48 is too extensive and confusing).

Report statistical power to support suficient sample size.

Please follow MDPI Reference List and Citation Style.

Specific comments:

Line 161 consider defining “muscle pain” or more specific meaning rather than perceived wellbeing. Although both are subjective perceptions, they do not represent the same thing.

Author Response

I would like to commend the authors on the revision of the manuscript and thank them for the point-by-point response. The paper is greatly improved. There are a couple of minor points that could do with further clarification.

Thank you once again for the careful review of our manuscript.

The manuscript should be reviewed by a native English speaker to check for syntax and flow (Example Line 45-48 is too extensive and confusing).

We have once again thoroughly revised the manuscript with special reference to syntax and flow. As a result we also performed some minor corrections in the discussion section (i.e. repeating information). Please see our amendments in red.

Report statistical power to support suficient sample size.

The statistical power has now been reported.

Please follow MDPI Reference List and Citation Style.

The reference style has been changed.

Specific comments:

Line 161 consider defining “muscle pain” or more specific meaning rather than perceived wellbeing. Although both are subjective perceptions, they do not represent the same thing.

Thanks for pointing this out. We agree with your suggestion and have modified this throughout the manuscript.